# Polygyny and intimate partner violence among married women: Sub-national estimates from a cross-sectional study in the Democratic Republic of the Congo

Zacharie Tsala Dimbuene [1]*, Bright Opoku Ahinkorah[2], Dickson Abanimi Amugsi[3]

**1** School of Population and Development Sciences, University of Kinshasa, Kinshasa, Democratic Republic of the Congo, **2** Faculty of Health, School of Public Health, University of Technology Sydney, Sydney, Australia, **3** African Population and Health Research Center, Nairobi, Kenya

☯ These authors contributed equally to this work.
* zacharie.tsala.dimbuene@gmail.com, zacharie.dimbuene@unikin.ac.cd

## Abstract

Intimate partner violence (IPV) is a global issue with several social and health consequences. Global estimates indicate that one-third of women have experienced lifetime IPV. In 2013, sub-Saharan Africa recorded the highest rates of IPV. Furthermore, previous research showed that polygyny is positively associated with IPV. This study examined associations between polygyny and IPV in Democratic Republic of the Congo (DRC) with a special attention to geographical variations. The paper used a subsample of 3,749 married women from 2013–2014 Demographic and Health Survey (DHS) in the DRC. Univariate and multivariable logistic regression was conducted to test statistical significance between polygyny and IPV and $p < 0.05$ was considered statistically significant. Findings showed spatial variations for polygyny and the three types of IPV. Overall, 19.0% of married women were in polygynous unions. This percentage ranged from 5.7% in North Kivu to 29.4% in Kasai occidental. In the last 12 months, 28.6%, 27.8%, and 19.6% of married women reported physical, emotional, and sexual violence, respectively, while 43.2% reported any form of IPV. IPV rates ranged from 18.1% in Kongo central to 58.3% in Kasai occidental. Net of controls, women in polygynous unions living Bandundu [AOR = 2.16, 95%CI = 1.38–3.38], Katanga [AOR = 1.78, 95%CI = 1.09–2.89], North Kivu [AOR = 6.22, 95%CI = 1.67–23.22], and South Kivu [AOR = 2.79, 95%CI = 1.03–7.54] had higher rates of IPV than their counterparts in monogamous unions. Spatial analyses showed that Kasai Occidental had the highest rates of IPV. Overall, being in polygynous increased significantly the odds of IPV. Programmatically, policymakers and stakeholders need to devise more effective policies and IPV interventions targeting polygynous families in DRC to achieve Sustainable Development Goals (SDG) 5.2, that aimed to eliminate all forms of violence against women and girls by 2030.

**Data Availability Statement:** Data are available from https://dhsprogram.com/data/available-datasets.cfm upon request.

**Funding:** The authors received no specific funding for this work.

**Competing interests:** The authors have declared that no competing interests exist.

## Introduction

Gender-based violence (GBV), referred to as a "set of harmful behaviours mostly against women and girls" [1], has increasingly attracted attention of researchers and policymakers over the past three decades [2,3]. This culminated in the adoption of Sustainable Development Goals (SDG) by the United Nations in 2015 [4], especially SDG 5.2 aimed at eradicating all forms of violence against women and girls by 2030 [4]. GBV is a violation of basic human rights; the most prevalent form of abuse and violence usually happens within intimate relationships. This type of violence is termed "intimate partner violence (IPV)". IPV encompasses physical, emotional and sexual violence [5] and it is linked to numerous social and health consequences. These include, loss of pregnancy through stillbirths and miscarriages, sexually transmitted infections [6], high levels of depression, post-traumatic stress disorder, psychological distress, and suicidal thoughts [7–9]. Admittedly, IPV is a multifaceted problem in almost all societies, [10] with few exceptions [1]. For instance, research suggests that domestic violence was virtually absent in small-scale societies like the Wape of Papua New Guinea, partly because the Government promotes egalitarian rules between women and men [11,12].

Worldwide, the prevalence of IPV was estimated at 30% [13] while it was estimated at 36% in sub-Saharan Africa (SSA) [14]. Significant variations were observed across regions, with SSA being severely affected. These high levels of IPV prevalence could hinder the subregion's ability to achieve the SDGs by 2030, unless measures necessitating urgent action are implemented within countries and across SSA [15]. Clearly, there is evidence of uneven distribution of IPV worldwide, across regions and countries, with developing countries being the most affected [13]. Indeed, data suggest that lifetime physical violence ranged from 12.9% in Japan City to 61.0% in the Peru province.

Efforts have been made by researchers and key stakeholders to address inequalities of GBV within world's regions [5,10,16]. This was an important milestone towards a better understanding of the geographical distribution of IPV within SSA region. However, these efforts ignored the disparities of IPV within countries; yet a better understanding of the geographical distribution of IPV within countries could help governments and stakeholders to prioritize their actions and design more localized effective interventions and programs to eradicate IPV while paying a special attention on polygynous families.

To further our understanding on IPV worldwide, researchers have devoted much time to find the putative correlates of IPV. Drawing on Bronfenbrenner's ecological framework, some studies posited that correlates of IPV are found at individual/personal, household, and community levels [5,17–19]. At individual and household level, age at first marriage, age differences between spouses, education, employment status, poverty proxied by household wealth index, number of cowives, and women's empowerment and decision-making, are among others, putative factors that drive IPV [16,20–27]. At community level, place of residence, poverty, gender norms and discrimination, wars and armed conflict were found to correlate with IPV [5,8,16,24,25,28–32].

While those studies provided further understanding on IPV and its correlates, others had paid special attention to the interlinkages between polygyny, the practice of one man being married to more than one wife at the same time [33], and IPV in SSA [10,33–39]. Overall, these studies found mixed results between polygyny and IPV. Some of these studies found that women in polygynous unions experienced higher rates of IPV compared with women in monogamous unions. For instance, in a multilevel modeling study in Mozambique, senior wives were at higher risks of IPV compared with younger wives [39]. Likewise, a multi-country study found that, net of controls, polygyny was positively associated with IPV in countries such as Angola, Burundi, Ethiopia, Uganda, Malawi, Mozambique, Zambia, and Zimbabwe.

However, in Cameroon and Nigeria, polygyny was negatively associated with IPV [10]. Using nationally representative data from 16 countries, this study provides insights on intricacies between polygyny and IPV at the national level and therefore masking the disparities on IPV within countries. Previous research clearly suggests that relationships between societal structure and IPV are very complex. While patrilineal societies may reinforce power imbalances that contribute to IPV, matrilineal societies can promote female empowerment and community support, potentially reducing violence. However, other factors (e.g., socio-economic conditions, legal frameworks, and cultural attitudes toward violence) could influence these dynamics. Understanding these nuances is crucial for developing effective prevention and intervention strategies.

The present study mimics multi-country studies addressing the interlinkages between polygyny and IPV in SSA [10]. To investigate the heterogeneity of IPV in DRC, this study adopted a sub-national perspective, taking the province as the unit of interest to unpack potential disparities within the country. The main hypothesis tested in this paper is that "polygyny is positively associated with IPV; however, provincial variations of these effects do exist". Findings are expected to provide more programmatic implications for governments and stakeholders to devise more effective and efficient interventions and programs to tackle the IPV issue and accelerate the achievement of SDG 5.2.

## Methods

### Study setting

The Democratic Republic of the Congo (DRC) offers a unique case to study the heterogeneity of IPV within a country for many reasons. The decades of war have reinforced patriarchal structures, and dangerous conditions for women within and outside the home [40]. The DRC was named the 'rape capital of the world' [41]. Therefore, combating GBV in the DRC at the individual, family, and community levels is of chief importance for organizations dedicated to supporting the SDGs of gender equality (Goal 5), health and wellbeing of women and girls (Goal 3) as well as peaceful, just, and inclusive societies (Goal 16). Further, the DRC is a mix of traditional patrilineal and matrilineal societies [42] despite the increasing urbanization observed in the country over the last three decades [43]. This feature of DRC's communities is essential to understand the geographical distribution of IPV in the country since a set of theories (e.g., feminist perspective) explain IPV as a consequence of patriarchy, the male dominance over women in many societies around the world [26,31,44]. Except for the province of Kinshasa, the Capital City of the country, the DRC is firstly made up of matrilineal societies (South Bandundu and Kongo central). However, a latent form of patriarchy still governs these provinces because men have more political power, moral authority and control of property than women. The remainder of the provinces (North Bandundu, Orientale, North Kivu, South Kivu, Katanga, Kasai Occidental and Kasai Oriental) are patrilineal societies with a clear male dominance. Previous studies showed that inadequate attention to the social and cultural factors that perpetuate and maintain GBV undermines the efficacy of current and future interventions [5]. Therefore, the present study also provides new insights into the mapping of polygyny and IPV in the country.

On legal aspects regarding violence against women and girls, the DRC ratified in 2008 the Protocol to the African Charter on Human and Peoples' Rights on the Rights of Women in Africa, the Maputo Protocol [45]. Furthermore, the DRC's Constitution clearly states that 'all persons have the right to life and to physical integrity' (Art. 16, 2011) [46]. However, there has been less progress in implementation. Indeed, there is no comprehensive law addressing violence against women and girls, even though existing legislation provides protections against

specific forms of violence against women and girls, including rape and sexual harassment. In her attempt to fight violence against women and girls, DRC's government developed a National Strategy against gender-based violence in 2009, established the National Agency for Eliminating Violence against Women and Adolescent and Very Young Girls (AVIFEM), and created a National Fund for the Promotion of Women and the Protection of Children (FONA-FEN) [47,48]. The National Strategy against gender-based violence (SNVBG) was built on five pillars: (*i*) fighting impunity, (*ii*) ensuring protection and prevention, (*iii*) undertaking security sector reforms. (*iv*) ensuring multi-sectoral assistance and (*v*) data and mapping. While all these initiatives are operational, they often experience financial challenges. Finally, violence against women and girls is endemic due to numerous factors, including discriminatory attitudes towards women, outdated customs, weak legal and judicial systems, culture of silence of victims and impunity of perpetrators [47].

## Data source

The paper utilizes data from the Demographic and Health Survey (DHS) conducted in the Democratic Republic of the Congo in 2013–2014 (DRC-DHS 2013–14) [49]. This is a nationally representative survey, using a two-stage sampling design, which collected information on households, women and men of reproductive ages, anthropometric measures, contraception and family planning, among others [49]. The first stage involved the selection of sample points or clusters from an updated master sampling frame constructed in accordance with DRC's administrative division in 26 provinces or domains. These domains were further stratified into urban and rural areas. From the urban areas, neighbourhoods were sampled from cities and towns whereas villages and chiefdoms were sampled for rural areas. The clusters were selected using systematic sampling with probability proportional to size. Household listing was then conducted in all the selected clusters to provide a complete sampling frame for the second stage selection of households.

The second stage of selection involved the systematic sampling of the households listed in each cluster, and households to be included in the survey are randomly selected. The rationale for the second stage selection is to ensure adequate numbers of completed individual interviews to provide estimates for key indicators with acceptable precision. All women aged 15–49 years in selected households were eligible to participate in the survey if they were either usual residents of the household or visitors present in the household on the night before the survey. This paper used data from married women in the individual record file to construct the outcome and independent variables. The analysis in the present study were restricted to married women because polygynous unions are easier to define in the context of marriage. The final sample consisted of 3,749 married women.

## Variables measurement

**Outcomes.** For the present study, the outcome variable was intimate partner violence (IPV), including physical, emotional, and sexual violence [10,19]. The three types of IPV were derived from the optional domestic violence module. In this module, questions about domestic violence in the last 12 months were asked, based on a modified version of the conflict tactics scale [50,51]. Questions for each type of IPV and responses are summarized in Table 1 below.

## Key independent variable

This study was interested in the associations between polygyny and IPV in the DRC. In the DHSs, married women reported the number of other wives that the husband had [10,33,52]. Similar to previous studies, women who indicated that their husbands had no other wives

**Table 1. Questions and responses about intimate partner violence.**

| Types of IPV and questions | Responses | Operational definition |
|---|---|---|
| **Physical violence (7 items)** | | |
| 1. Husband ever pushed, shook, or threw something<br>2. Husband slapped<br>3. Husband punched her with his fist or something harmful<br>4. Husband kicked or dragged<br>5. Husband strangled or burnt<br>6. Husband threatened her with a knife, gun, or other weapons<br>7. Husband twisted arm or pulled hair | Responses included 0 "Never"; 1 "Often"; 2 "Sometimes"; and 3 "Yes, but not in the last 12 months" | The item was recorded 0 "NO" if wife reported "Never" or "Yes, but not in the last 12 months" and 1 if wife reported "Often" or "Sometimes". The new variable ranged from 0 to 7 |
| **Emotional violence (3 items)** | | |
| 1. Husband humiliated<br>2. Husband threatened to harm<br>3. Husband insulted or made feel bad | Responses included 0 "Never"; 1 "Often"; 2 "Sometimes"; and 3 "Yes, but not in the last 12 months" | The item was recorded 0 "NO" if wife reported "Never" or "Yes, but not in the last 12 months" and 1 if wife reported "Often" or "Sometimes". The new variable ranged from 0 to 3 |
| **Sexual violence (3 items)** | | |
| 1. Husband ever physically forced wife into unwanted sex<br>2. Husband ever forced wife into other unwanted sexual acts<br>3. Respondent has been physically forced to perform sexual acts she didn't want to | Responses included 0 "Never"; 1 "Often"; 2 "Sometimes"; and 3 "Yes, but not in the last 12 months" | The item was recorded 0 "NO" if wife reported "Never" or "Yes, but not in the last 12 months" and 1 if wife reported "Often" or "Sometimes". The new variable ranged from 0 to 3 |

were considered being in monogamous marriages. In contrast, those who indicated that husbands had at least one or more other wives were considered living in polygynous marriages. Therefore, the variable 'polygyny' used in this study is a dichotomous variable taking the value "1" if the woman was living in polygynous marriages and "0" if otherwise. Other studies have moved further to understand the paths of influence of polygyny on IPV. In Mozambique, a study on the associations between polygyny and IPV conceptualized polygyny taking into account the rank of women in polygynous unions and coresidence [39]. This study found, net of controls, that senior wives reported higher rates of IPV. Additionally, women in polygynous unions living away from their cowives reported higher rates of IPV compared to women coresiding with cowives and women in monogamous unions. In the current study, additional analyses were performed to check the associations between the number of cowives and IPV.

## Controls

The literature suggests that the sociodemographics that define risk groups for IPV [53–55], and the causes of IPV are more complex than generally stated. Jewkes showed that women who are more empowered educationally, economically and socially are usually most protected against IPV [53]. In this study, level of education was coded as no education, primary, secondary, and higher. Other studies have focused on urban residence and showed how urban residence could explain rates of IPV in SSA [24,56]. A meta-analysis on IPV in SSA showed that women living in rural areas experienced higher rates of IPV than urban counterparts. In this study, place of residence was coded '1' if the woman resided in urban areas, and '0' otherwise. Poverty was also found to be associated with IPV: women in better-off households experienced lower rates of IPV [57]. Poverty in this study was proxied by the household wealth index (HWI). The methodology to derive HWI from DHSs data has been published elsewhere

[58,59]. HWI in the original dataset was categorized as 'poorest', 'poorer', 'middle', "richer", and 'richest'. In this study, the original categorization of wealth quintile as used in the DHS was adopted. The index of media exposure was created from three variables: the frequency of watching television, listening to radio, or reading newspapers/magazines. Responses to these variables were '0' if respondent reported 'not at all', '1' for 'less than once a week', and '2' for 'at least once a week'. In this paper, responses were recorded into '0 = No' for 'not at all' and '1 = Yes' for 'less than once a week' and 'at least once a week'. Finally, a dichotomous variable was created from a composite of exposure to the three media sources and defined as "0 = No" for married women who scored '0' on the three items and '1 = Yes' if women's score was higher or equal to '1'. Previous studies also found that attitudes towards violence were associated with IPV [35,60–63]. The variable 'justification of violence' was derived from questions asking married women if it is justified for a husband to beat his wife for the following reasons: (*i*) burning food, (*ii*) arguing with him, (*iii*) going out without telling him, (*iv*) neglecting the children, and (*v*) refusing to have sexual intercourse with him. A binary variable was created from these five reasons to reflect the attitudes towards wife beating. Justification of violence was therefore coded as '0 = No' if a woman disagreed with the five reasons and '1 = Yes' if she agreed to at least one of these reasons.

## Analytical strategy

The analyses first evaluated the prevalence of polygyny and IPV in the last 12 months among married women at national and sub-national levels using spatial variations. Second, associations between polygyny and IPV, as well as its components were examined. Statistical significance was tested at national and sub-national levels using Pearson's chi-square tests. Finally, the association between polygyny and IPV at national and sub-national levels was assessed using bivariate and multivariable binary logistic regression models. Unadjusted odds ratio (UOR) and adjusted odds ratios (AOR), along with 95% confidence intervals (CIs) were reported. Given the complex sampling design in the DHSs, analyses used the command **svy**, and the option **subpop** in STATA SE 15 to obtain unbiased estimates. To provide good estimates, the entire sample was kept in the dataset, using the option **subpop** to select the sub-sample of interest.

**Preliminary analyses.** Multicollinearity tests and the statistical significance of the associations between IPV, key independent variables and controls were examined. These methodological issues are extensively discussed in the literature [64,65]. Using variance tolerance, known as Variance Inflation Factor (VIF), the tests revealed no multicollinearity problems. Overall, VIF was 1.35 and it ranged from 1.02 to 1.81.

**Goodness-of-fit of the models and the influence of the outliers.** Another issue discussed in multivariable logistic regression is the goodness-of-fit of the estimated models: To what extent do models significantly fit the data? Tests used include log-likelihood, test of Hosmer-Lemeshow, Pearson's Chi-square of the model and the Receiver Operating Characteristic (ROC) curve. The influences of outliers on the estimates were examined using a plot of the residuals and predicted probabilities of the outcome to check for covariate patterns and over-dispersion. Residuals with absolute values above 1 indicate a problematic covariate pattern that can undermine the goodness-of-fit of the models. However, the plots depicted no residual values above 1 or overdispersion issues.

## Ethics statement

The DHS obtained ethical clearance from the recognised Ethical Review Committees/Institutional Review Boards of the Democratic Republic of the Congo as well as the Institutional

Review Board of ICF International (USA) before the surveys were conducted. Written informed consent was obtained from the women before participation. The authors of this paper sought and obtained permission from the DHS programme to use the data. The data was completely anonymized; therefore, the authors did not seek further ethical clearance before their use.

## Results

### Descriptive results

The final sample consisted of 3,749 married women and was unevenly distributed by place of residence and province. Indeed, 29.9% of married women were urban residents, and geographically, the percentage distribution ranged from 2.4% in Kongo central to 15.2% in Kasai Oriental (Table 2).

### Prevalence of polygyny

The results showed that 19.0% of married women in the final sample were in polygynous unions with significant spatial variations (Fig 1). The percentage of married women in polygynous unions varied from 5.7% in North Kivu to 29.4% in Kasai occidental. Also, more than one-fifth of married women were in polygynous unions in Equateur (22.0%), Bandundu (22.5%), and Kasai oriental (26.9%).

### Prevalence of intimate partner violence

Figs 2–5 display the spatial distribution of physical, emotional, and sexual violence, and the overall intimate partner violence. Results indicated that, 28.6% of married women reported having experienced physical violence in the last 12 months preceding the survey. It is followed by emotional violence (27.8%) while sexual violence was at 19.6%.

From Fig 2, results revealed geographical variations of physical violence among married women. It ranged from 10.6% in North Kivu to 42.4% in Kasai Occidental. Similar patterns were observed for emotional violence (Fig 3) and sexual violence (Fig 4). The province of Kasai occidental had the highest prevalence emotional (38.5%) and sexual violence (29.6%).

**Table 2. Percentage distribution of married women, and percentage of married women in polygynous women per province.**

| Province of residence | N | Percentage | Percentage of married women in polygynous unions (*) |
|---|---|---|---|
| Kinshasa | 157 | 4.2 | 6.0 |
| Bandundu | 502 | 13.4 | 22.5 |
| Kongo Central | 89 | 2.4 | 6.1 |
| Equateur | 514 | 13.7 | 22.0 |
| Kasai Occidental | 443 | 11.8 | 29.4 |
| Kasai Oriental | 570 | 15.2 | 26.9 |
| Katanga | 562 | 15.0 | 17.4 |
| Maniema | 248 | 6.6 | 14.3 |
| North Kivu | 170 | 4.5 | 5.7 |
| Orientale | 276 | 7.4 | 11.8 |
| South Kivu | 218 | 5.8 | 13.7 |
| Total | 3,749 | 100.0 | **19.0** |

Note: (*) These figures represent the percentages of married women in polygynous union per province, and the entire sample.

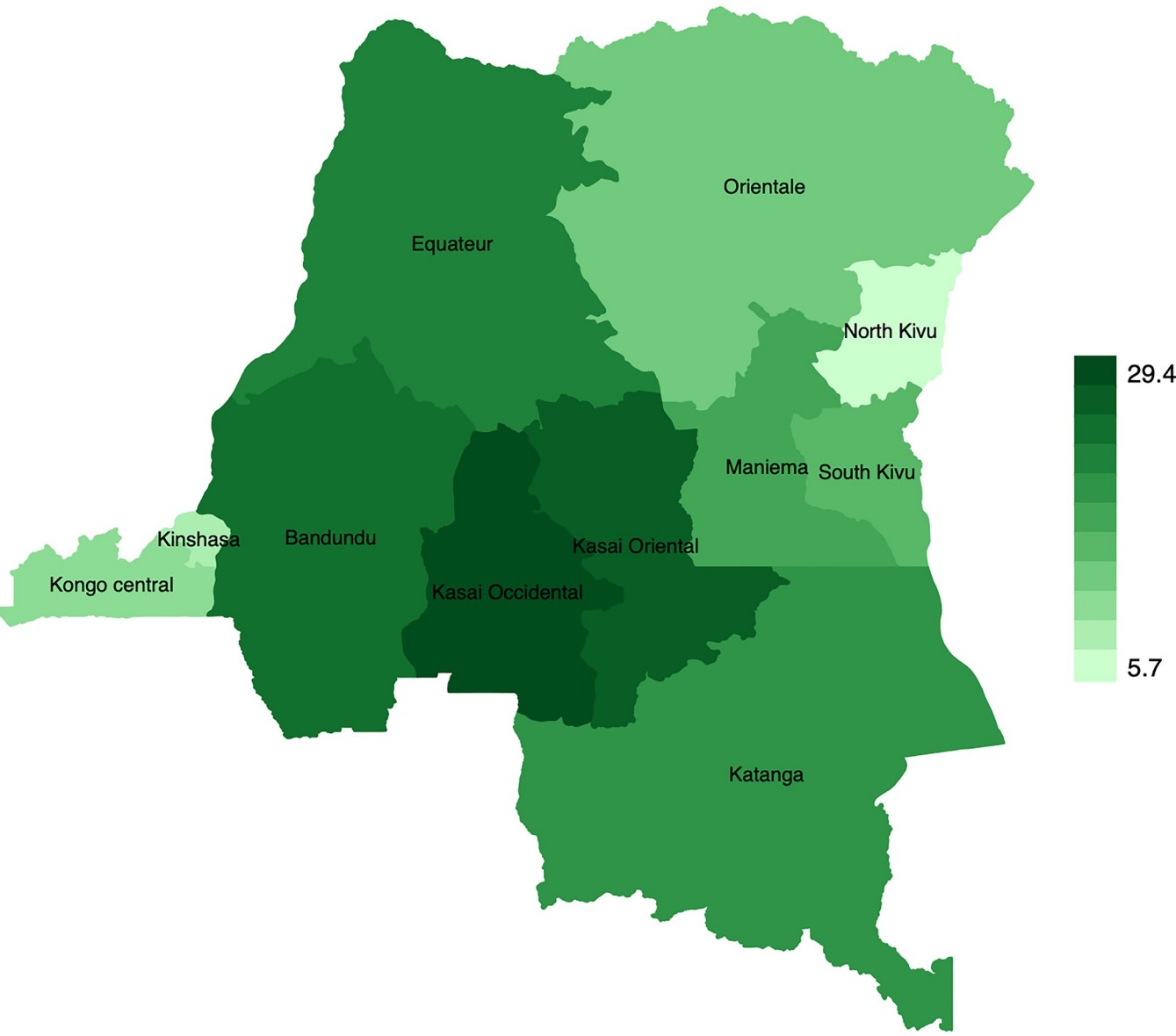

**Fig 1. Percentage distribution of polygynous unions in the Democratic Republic of the Congo by province.** Note: Shapefiles to build maps were downloaded from https://www.idhsdata.org/idhs/gis.shtml.

Overall, findings showed that 43.2% of married women were exposed to IPV in the last 12 months preceding the survey with significant spatial variations (Fig 5). Again, married women in Kasai occidental experienced higher rates of IPV (58.3%). In contrast, married women from Kongo central had the lowest rates of IPV with 18.1%. The rates of IPV were also prevalent among married women in Kasai oriental (52.4%), Bandundu (48.8%), Equateur (44.6%), South Kivu (41.2%) and Katanga (40.4%).

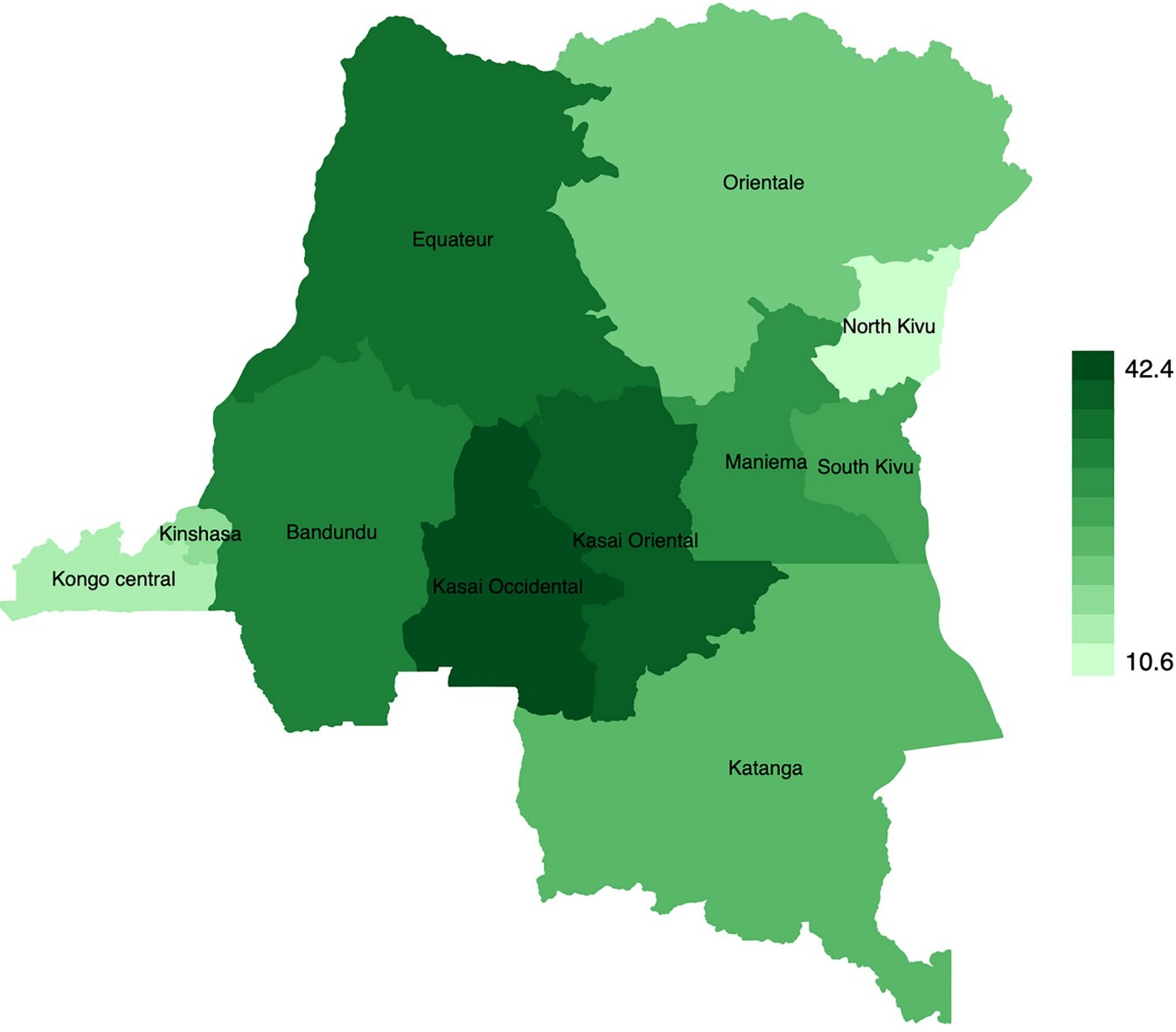

## Pct. Physical violence
### Democratic Republic of the Congo, 2013-2014

**Fig 2. Percentage distribution of physical violence in the Democratic Republic of the Congo by province.** Note: Shapefiles to build maps were downloaded from https://www.idhsdata.org/idhs/gis.shtml.

### Polygynous unions and physical, emotional, sexual, and intimate partner violence

Table 3 below shows the distribution of the different forms of IPV across women's polygyny status along with statistical significance. The rates of IPV are higher among women in polygynous unions, irrespective of the type of IPV and the overall IPV in the last 12 months. For

# Pct. Emotional violence
## Democratic Republic of the Congo, 2013-2014

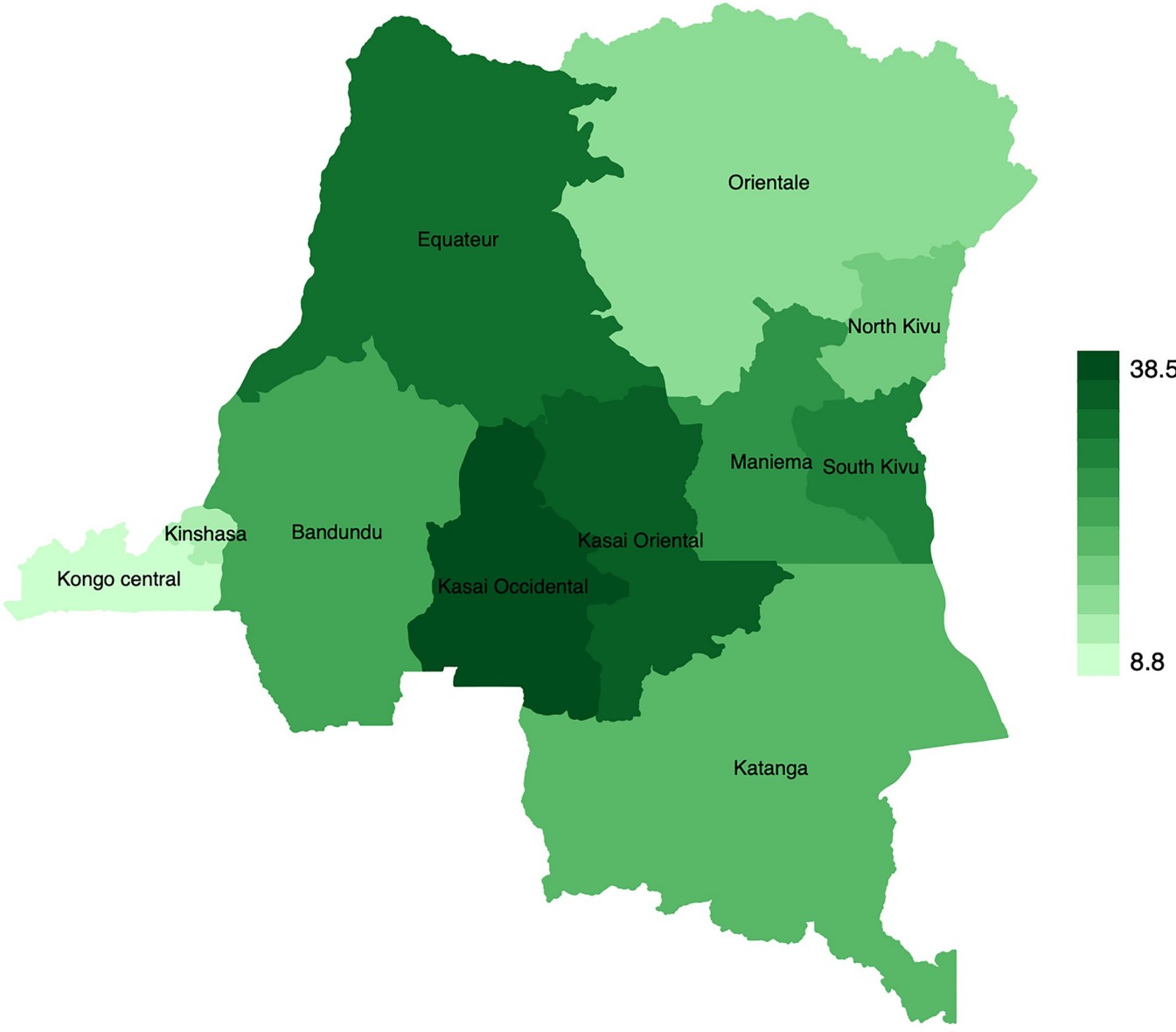

**Fig 3. Percentage distribution of emotional violence in the Democratic Republic of the Congo by province.** Note: Shapefiles to build maps were downloaded from https://www.idhsdata.org/idhs/gis.shtml.

instance, it was estimated at 54.0% among married women in polygynous unions compared with 40.6% among those in monogamous unions. The difference was statistically significant ($p < 0.001$).

The rates of physical, emotional, and sexual violence among married women in polygynous unions in the last 12 months at national level were 39.8%, 37.8% and 22.8%, respectively. The corresponding figures for married women in monogamous unions were lower at 26.0%, 25.5%

# Pct. Sexual violence
## Democratic Republic of the Congo, 2013-2014

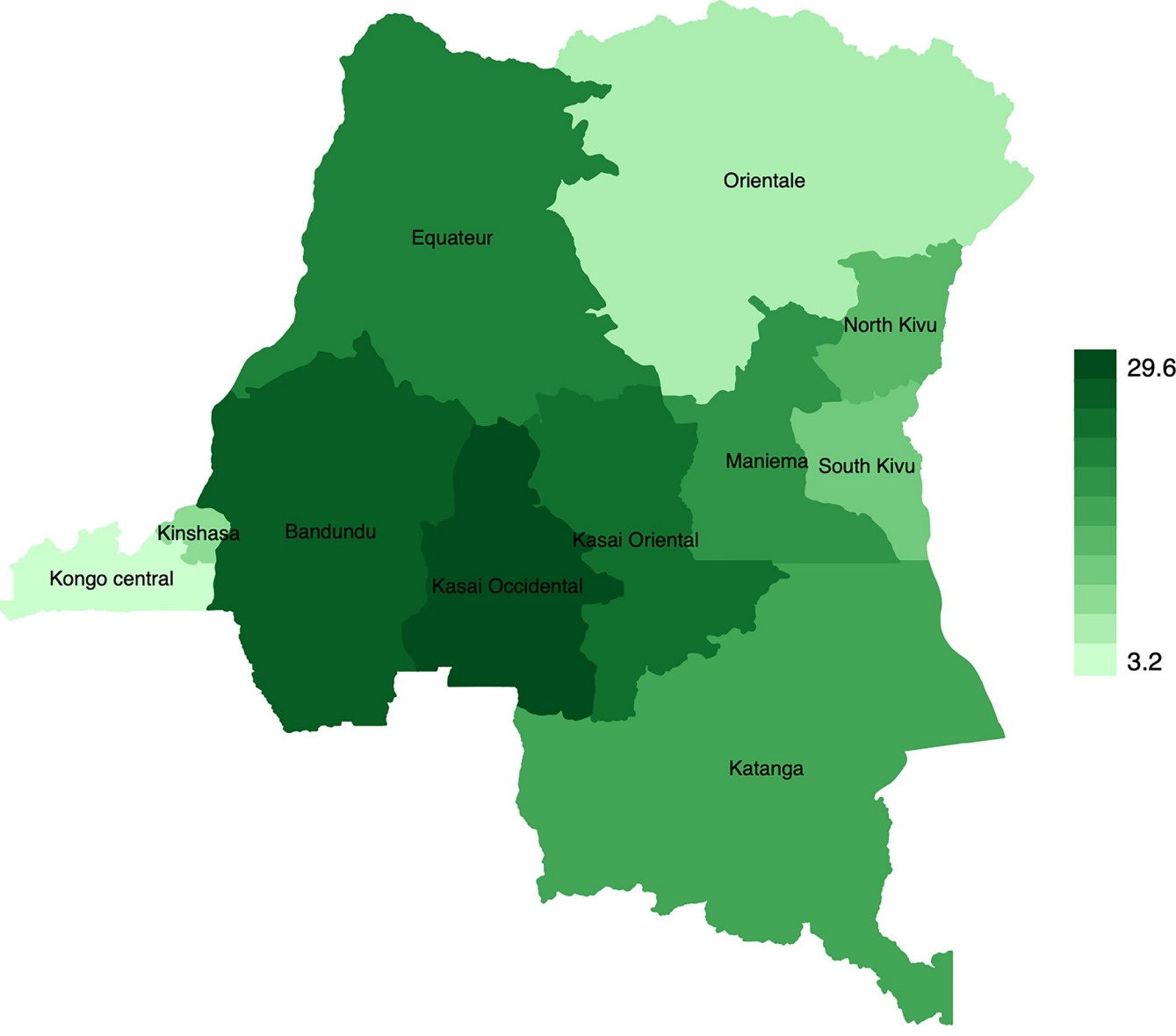

**Fig 4. Percentage distribution of sexual violence in the Democratic Republic of the Congo by province.** Note: Shapefiles to build maps were downloaded from https://www.idhsdata.org/idhs/gis.shtml.

and 18.9%, respectively. For each type of IPV, findings indicated statistical significance ($p < 0.001$).

The analyses revealed the same pattern at sub-national level. For physical violence, the most notable difference was observed among married women in polygynous unions from Kongo central with rates of physical violence being almost 5 times higher than of their counterparts in monogamous unions ($p < 0.000$). Similar pattern was observed for emotional violence in

# Pct. Intimate partner violence
## Democratic Republic of the Congo, 2013-2014

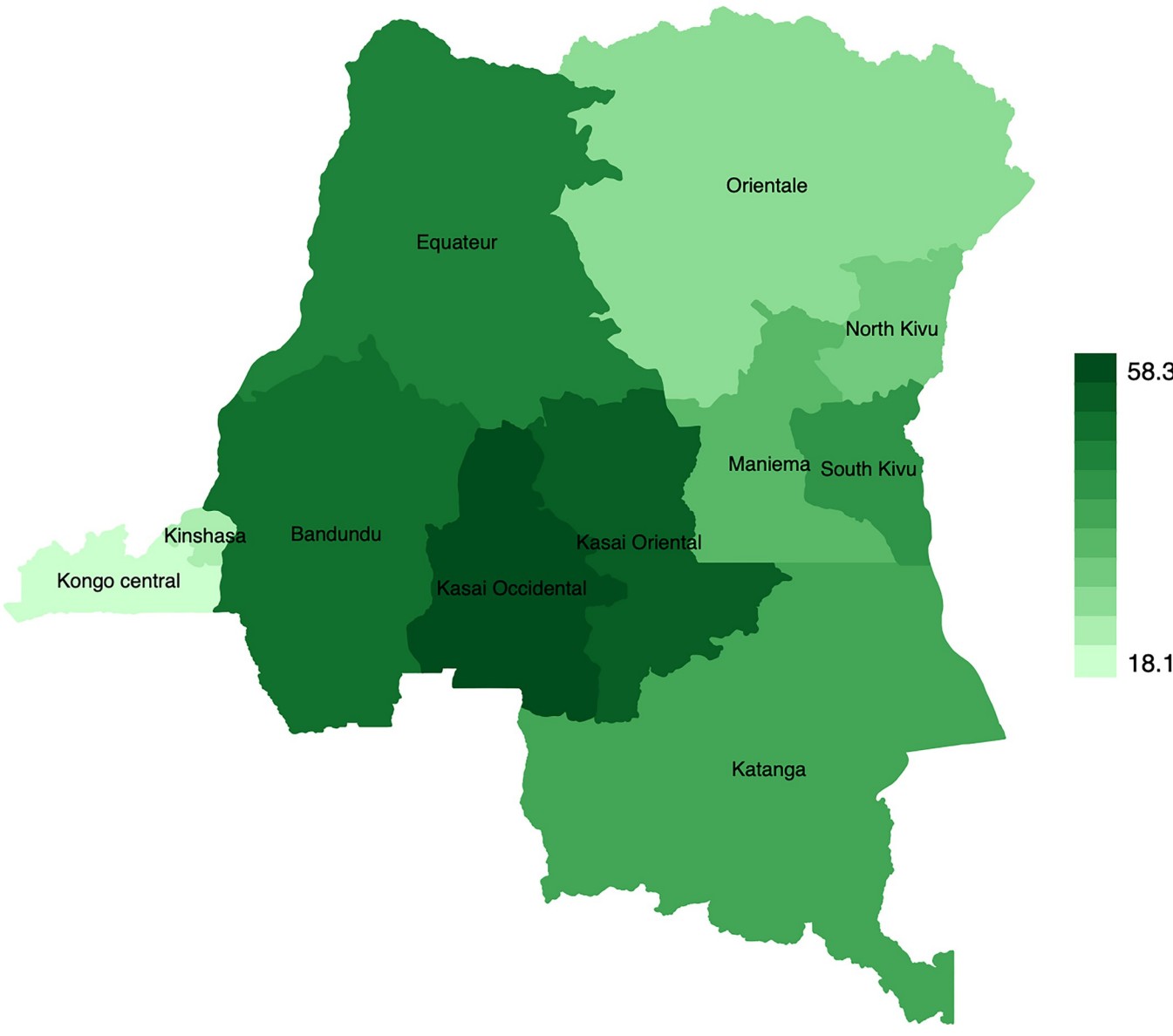

**Fig 5. Percentage distribution of intimate partner violence in the Democratic Republic of the Congo by province.** Note: Shapefiles to build maps were downloaded from https://www.idhsdata.org/idhs/gis.shtml.

North Kivu where the rates of emotional violence among women in polygynous marriages were higher (66.4%) compared with women in monogamous marriages (17.5%). Sexual violence showed different patterns in some provinces (Equateur, Kasai occidental, Katanga, and South Kivu) where the rates of sexual violence were instead higher among women in monogamous marriages.

**Table 3. Polygyny and intimate partner violence (%) in the last 12 months by province in the Democratic Republic of the Congo.**

| Province | Physical violence | | | Emotional violence | | | Sexual violence | | | Intimate Partner Violence | | |
|---|---|---|---|---|---|---|---|---|---|---|---|---|
| | Monogamous | Polygynous | p-value | Monogamous | Polygynous | p-value | Monogamous | Polygynous | p-value | Monogamous | Polygynous | p-value |
| Kinshasa | 13.8 | 25.7 | 0.299 | 16.5 | 30.1 | 0.223 | 5.7 | 27.7 | 0.005 | 24.1 | 58.8 | 0.019 |
| Bandundu | 30.3 | 44.3 | 0.003 | 20.9 | 45.5 | 0.000 | 25.6 | 33.5 | 0.093 | 45.6 | 59.5 | 0.001 |
| Kongo Central | 10.3 | 44.8 | 0.000 | 8.3 | 16.4 | 0.461 | 3.4 | 0.0 | 0.753 | 16.4 | 44.8 | 0.001 |
| Equateur | 31.6 | 41.1 | 0.112 | 28.7 | 34.9 | 0.306 | 22.8 | 12.8 | 0.126 | 43 | 50.5 | 0.265 |
| Kasai Occidental | 41.8 | 44.1 | 0.706 | 37.0 | 42.1 | 0.349 | 30.8 | 26.7 | 0.485 | 58.9 | 56.7 | 0.757 |
| Kasai Oriental | 33.5 | 39.6 | 0.269 | 37.7 | 36.2 | 0.885 | 21.4 | 28.5 | 0.247 | 51.5 | 54.6 | 0.799 |
| Katanga | 22.5 | 30.3 | 0.205 | 23.5 | 30.8 | 0.160 | 18.4 | 14.7 | 0.357 | 39.2 | 46.4 | 0.260 |
| Maniema | 28.1 | 35.0 | 0.414 | 25.6 | 36.2 | 0.358 | 18.4 | 20.8 | 0.686 | 38.4 | 45.6 | 0.424 |
| North Kivu | 10.7 | 15.6 | 0.640 | 17.5 | 66.4 | 0.001 | 17.8 | 18.9 | 0.920 | 31.3 | 71.1 | 0.021 |
| Orientale | 15.9 | 29.1 | 0.053 | 18.6 | 29.0 | 0.236 | 5.8 | 14.8 | 0.099 | 25.1 | 35.9 | 0.257 |
| South Kivu | 23.0 | 52.4 | 0.002 | 27.7 | 29.8 | 0.859 | 12.9 | 3.4 | 0.066 | 37.9 | 61.8 | 0.034 |
| **All women** | **26.0** | **39.8** | **<0.001** | **25.5** | **37.9** | **<0.001** | **18.9** | **22.8** | **<0.001** | **40.6** | **54.0** | **<0.001** |

Notes: Analyses were done separately for each province. Cross-tabulation was done between "polygynous" status (coded 0 "No" for women in monogamous unions and 1 "Yes" for women in polygynous unions) and intimate partner violence (or each type of violence). Statistical significance was tested using Pearson's chi-square and corresponding p-values were reported in Table 3.

## Bivariate and multivariate results from logistic regression models

This section summarizes the associations between polygyny and intimate partner violence via unadjusted odds ratios (UOR) in Model 1 and adjusted odds ratios (AOR) in Model 2, controlling for other covariates (Table 4) for the entire sample and by province.

The likelihood of reporting IPV in the entire sample was higher among women in polygynous marriages. In the bivariate analysis (Model 1, Table 4), women in polygynous marriages had higher odds of reporting IPV in the last 12 months [UOR = 1.71, 95%CI = 1.316–2.224]. Net of controls (Model 2, Table 4), being in polygynous marriages increased the odds of reporting IPV in the last 12 months [AOR = 1.64, 95%CI = 1.24–2.17]. In examining the province-specific effects of polygyny on IPV (Model 1, Table 4), results showed that overall, being in polygynous unions increased the likelihood of reporting IPV, except in Kasai occidental. However, only five provinces reached statistical significance. These include Kinshasa [UOR = 4.48, 95%CI = 1.19–16.9], Bandundu [UOR = 1.75, 95%CI = 1.28–2.40], Kongo central [UOR = 4.15, 95%CI = 1.87–9.22], North Kivu [UOR = 5.40, 95%CI = 1.14–25.49] and South Kivu [UOR = 2.65, 95%CI = 1.06–6.64]. When controls are introduced in the estimations (Model 2, Table 4), a similar pattern was found. However, while the associations between polygyny and IPV were no longer significant in Kinshasa, they became significant in Katanga. The results were as follow: Bandundu [AOR = 2.16, 95%CI = 1.38–3.38], Katanga [AOR = 1.78, 95%CI = 1.09–2.89], North Kivu [AOR = 6.22, 95%CI = 1.67–23.22], South Kivu [AOR = 2.79, 95%CI = 1.03–7.54], and marginally in Kongo central [AOR = 4.76, 95% CI = 0.92–24.65]. Additional analyses (*Not shown*) using the number of cowives as key independent variable while controlling with same variables are presented (See S1 Table).

**Table 4. Bivariate and multivariate estimated odds ratios between polygyny and intimate partner violence.**

| Province | Model 1 | | Model 2 | |
|---|---|---|---|---|
| | UOR | 95% CI | AOR | 95%CI |
| Kinshasa | 4.482** | (1.186–16.942) | 2.755 | (0.672–11.287) |
| Bandundu | 1.753*** | (1.280–2.401) | 2.161*** | (1.382–3.379) |
| Kongo Central | 4.147*** | (1.866–9.218) | 4.758* | (0.918–24.651) |
| Equateur | 1.351 | (0.792–2.305) | 1.521 | (0.802–2.886) |
| Kasai Occidental | 0.912 | (0.503–1.656) | 0.848 | (0.448–1.604) |
| Kasai Oriental | 1.136 | (0.421–3.063) | 1.103 | (0.401–3.035) |
| Katanga | 1.343 | (0.799–2.258) | 1.775** | (1.089–2.891) |
| Maniema | 1.344 | (0.634–2.847) | 1.151 | (0.392–3.382) |
| North Kivu | 5.396** | (1.142–25.489) | 6.221*** | (1.666–23.225) |
| Orientale | 1.674 | (0.676–4.144) | 2.125 | (0.844–5.353) |
| South Kivu | 2.650** | (1.058–6.640) | 2.786** | (1.030–7.535) |
| **All married women** | **1.711***** | **(1.316–2.224)** | **1.638***** | **(1.235–2.174)** |

95% Confidence interval of Unadjusted Odd Ratio (UOR) and Adjusted Odd Ratio (AOR) in parentheses. Model 2 controls for women's education, age, urban residence, household's wealth index, and attitudes towards intimate partner violence.

Significance level

*** p<0.01

** p<0.05

* p<0.1.

Source: DHS DRC 2013–14.

## Discussion

Previous research showed that IPV is a global social and health issue, given the related social and health consequences [6,13,53,66]. While global estimates indicated that overall, 30% had experienced lifetime IPV, the present study showed that the prevalence of IPV among married women in the DRC was higher than the global estimate, with 43.2% having experienced lifetime IPV. This study was specifically interested in the association between polygyny, a marriage practice that remains very common in most SSA societies, and IPV. Although previous research has made tremendous efforts to further our understanding of the relationship between polygyny and IPV, and woman's and child's health [10,33–37,39,52,67], findings from these studies mask the provincial variations of the effects of polygyny on IPV.

There are justifiable reasons to focus on the DRC. First, the country has been devastated by three decades of war and is struggling to implement a sustainable environment to fight violence against women and girls. Second, previous research showed that the quality of public institutions was worse in Central Africa than any other sub-region in SSA [68]. Third, the mix of patrilineal and matrilineal societies in the DRC makes it an interesting case study to see how the effects of polygyny on IPV can vary at sub-national level.

At the national level, findings were in line with previous studies [10,39,69]. Women in polygynous marriages were more exposed to IPV than their counterparts in monogamous marriages. At sub-national level, there is strong evidence of geographical disparities of IPV across provinces, irrespective of the form of IPV. The mapping exercise showed that the distribution of IPV and its different types of violence mimicked the spatial distribution of polygyny. For instance, IPV was more prevalent in provinces where the rates of polygyny were higher. These include Kasai occidental, Kasai oriental, and Bandundu, among others. While the provinces of Kasai occidental and Kasai oriental are patrilineal societies, Bandundu is a mix of patrilineal societies in the North and matrilineal societies in the South.

Bivariate and multivariate associations between polygyny and IPV clearly highlighted the 'heterogeneity of IPV'. Indeed, while studies at national level furthered our understanding of the effects of polygyny on IPV, they also masked subnational variations. Yet, understanding subnational variations of IPV through context-specific studies are important for the design of targeted interventions and programs to address violence against women and girls, which will contribute to the DRC effort to achieve SDGs 5.2 by 2030 [4]. Further, since GBV and its most common form, IPV is rooted in structural gender inequality and power imbalances in most societies, it is important to consider the specificities of local environments. This study provides insights into the variations of the effects of polygyny on IPV. This is particularly important in the context of limited resources on the one hand, and on the other hand, given the damage that IPV could cause to the development of a country [70], and the related economic costs of violence against women and girls [71].

### Strengths and limitations

This study used a nationally representative dataset to examine the association between polygyny and IPV. This is a strength because the original sampling design aimed at producing valid estimates at subnational level. However, it has some limitations concerning the measurement of key variables: polygyny and IPV. First, the measurement of "polygyny" or "polygynous marriages" might be problematic. It was captured with a single question about the number of "known" other wives. Second, women's sociodemographics were considered even though it included some factors at household and community levels. Third, previous research showed that the husband's characteristics are equally important to further our understanding of IPV in the communities [72]. However, this paper didn't include husband's characteristics. Fourth,

the study used cross-sectional data; therefore, findings should be interpreted in terms of associations, and no definite conclusion can be drawn. Fifth, IPV is captured through self-reported information in the last 12 months, with possibilities of under-and over-reporting of data. Sixth, this study used cross-sectional data, and as such, no definite conclusion could be drawn regarding causality between polygyny and IPV.

## Conclusion

Findings showed that IPV prevalence varies across provinces in the DRC, and so do the effects of polygyny on IPV at subnational level. Indeed, Kasai Occidental had the highest rates of IPV. Overall, being in polygynous increased significantly the odds of IPV; however, these associations varied across provinces. These findings have a policy and programmatic implications to the country. While the adverse effects of IPV on women's physical, emotional, and sexual health cannot be denied, policymakers and stakeholders should recognize that all provinces in the DRC are not equally affected, and some provinces require more attention than others if the country wants to achieve SDG 5.2. The fact that the prevalence of IPV mimicked the distribution of polygyny also calls for special attention. Policymakers and stakeholders should devise interventions and policies that target these intertwined issues taking into the specificities of each province. Therefore, there is an urgent need for localized interventions to tackle IPV. Specifically, IPV interventions aimed at equipping communities to offer more resources for women in polygynous relationships could be more effective. Future research should also explore additional factors, such as husband/partner's characteristics and community-level influences, to further our understanding of IPV drivers in different contexts. Finally, DRC's government should move towards a serious shift concerning the legal framework of violence against women and girls in the country.

## Supporting information

**S1 Checklist. STROBE checklist for *cross-sectional studies* adapted to our study.**
(DOCX)

**S1 Table. Estimated odds ratios of the number cowives on intimate partner violence in the Democratic Republic of the Congo by province.**
(DOCX)

## Acknowledgments

The authors wish to express their profound gratitude to The DHS Program, USA, for providing free access to the data. They also wish to acknowledge institutions of the Democratic Republic of the Congo that played critical roles in the data collection process. The authors also express their gratitude to Ms. Stella Kasura for editing previous versions of the manuscript.

## Author Contributions

**Conceptualization:** Zacharie Tsala Dimbuene, Bright Opoku Ahinkorah, Dickson Abanimi Amugsi.

**Data curation:** Zacharie Tsala Dimbuene.

**Formal analysis:** Zacharie Tsala Dimbuene.

**Methodology:** Zacharie Tsala Dimbuene, Dickson Abanimi Amugsi.

**Project administration:** Zacharie Tsala Dimbuene.

**Software:** Zacharie Tsala Dimbuene, Bright Opoku Ahinkorah.

**Supervision:** Zacharie Tsala Dimbuene.

**Validation:** Zacharie Tsala Dimbuene, Bright Opoku Ahinkorah.

**Visualization:** Zacharie Tsala Dimbuene.

**Writing – original draft:** Zacharie Tsala Dimbuene.

**Writing – review & editing:** Zacharie Tsala Dimbuene, Bright Opoku Ahinkorah, Dickson Abanimi Amugsi.

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
