## [Decision Letter · Decision Letter 0]

20 Nov 2023

PGPH-D-23-00057

Polygyny and intimate partner violence among married women: Sub-national estimates from a cross-sectional study in the Democratic Republic of the Congo

Dear Dr. Tsala Dimbuene,

Thank you for submitting your manuscript to PLOS Global Public Health. After careful consideration, we feel that it has merit but does not fully meet PLOS Global Public Health’s publication criteria as it currently stands. Therefore, we invite you to submit a revised version of the manuscript that addresses the points raised during the review process.

We look forward to receiving your revised manuscript.

Kind regards,

Abhijit Nadkarni

Academic Editor

Journal Requirements:

1. Figure 2 is currently divided into multiple panels in separate files. Please include all panels for each figure within a single file, with each panel in the image labelled alphabetically (A, B, C). References to panels within figures should include the figure number and panel letter: e.g., Fig 1A, Fig 1B, Fig 1C. Alternatively, you may wish to separate panels into individual figures, renaming them accordingly (Fig 1A as Fig 1, Fig 1B as Fig 2, etc.).

Please update figure legends and in-text figure citations accordingly.

2. Some material included in your submission may be copyrighted. According to PLOS’s copyright policy, authors who use figures or other material (e.g., graphics, clipart, maps) from another author or copyright holder must demonstrate or obtain permission to publish this material under the Creative Commons Attribution 4.0 International (CC BY 4.0) License used by PLOS journals. Please closely review the details of PLOS’s copyright requirements here: PLOS Licenses and Copyright. If you need to request permissions from a copyright holder, you may use PLOS's Copyright Content Permission form.

Potential Copyright Issues:

Figures 1 and 2: Please (a) provide a direct link to the base layer of the map (i.e., the country or region border shape) and ensure this is also included in the figure legend; and (b) provide a link to the terms of use / license information for the base layer image or shapefile. We cannot publish proprietary or copyrighted maps (e.g. Google Maps, Mapquest) and the terms of use for your map base layer must be compatible with our CC-BY 4.0 license. 

3. Please send a completed 'Competing Interests' statement, including any COIs declared by your co-authors. If you have no competing interests to declare, please state "The authors have declared that no competing interests exist". 

4. Please amend your detailed Financial Disclosure statement. This is published with the article. It must therefore be completed in full sentences and contain the exact wording you wish to be published.

5. We have noticed that you have uploaded Supporting Information files, but you have not included a list of legends. Please add a full list of legends for your Supporting Information files after the references list. 

Additional Editor Comments (if provided):

Reviewers' comments:

Reviewer's Responses to Questions

**Comments to the Author**

1. Does this manuscript meet PLOS Global Public Health’s publication criteria? Is the manuscript technically sound, and do the data support the conclusions? The manuscript must describe methodologically and ethically rigorous research with conclusions that are appropriately drawn based on the data presented.

Reviewer #1: Yes

2. Has the statistical analysis been performed appropriately and rigorously?

Reviewer #1: Yes

3. Have the authors made all data underlying the findings in their manuscript fully available (please refer to the Data Availability Statement at the start of the manuscript PDF file)?

Reviewer #1: Yes

4. Is the manuscript presented in an intelligible fashion and written in standard English?

Reviewer #1: Yes

5. Review Comments to the Author

Reviewer #1: This is a very thoroughly written manuscript on a very sensitive and important topic of intimate partner violence (IPV). The authors have demonstrated extensive knowledge in the subject matter of the paper providing important background literature and a well designed study. The methodology is well prepared and described and the extensive data and sample make the findings very interesting and generalization plausible. The statistical analysis is well conducted and well presented with appropriate tools and tables clearly presenting steps taken to perform the analysis. Again, the authors have taken time and effort to relate and compare the findings to other studies and clearly identified similarities and differences.

There are however, a few issues that I think will help improve the manuscript even better. The first has to do with some level of verbosity of the manuscript. It appears there is verbosity in presenting information such as the study setting, description of the study controls, and some portions of the discussion. Please, address this issue by providing summarized versions of the portions mentioned and other portions. The second issue is that there is a few instances of language or grammatical inconsistency and it will be great to run the manuscript through some form of editing professional or through Grammerly software. The third issue is about a statement you made on lines 450-451 in the conclusion about the lawfulness of men marrying more than one wife in DRC. I think this statement may raise concerns of bias, it may seem that polygyny is synonymous to IPV, which is not the case here. It is important to avoid such statements as it can edgy for some reasons, as IPV happens in all kinds of relationships.

Other than the few issues I have raised, I think this manuscript is very important and well prepared.

6. PLOS authors have the option to publish the peer review history of their article (what does this mean?). If published, this will include your full peer review and any attached files.

**Do you want your identity to be public for this peer review?** For information about this choice, including consent withdrawal, please see our Privacy Policy.

Reviewer #1: No

Reviewer 2 (Comments sent to Editor over email)

Language

• Language must be factual and non-evaluative. For example, the following types statements are not typical in academic articles:

1. To vehemently question the homogeneity of IPV within countries…

in the Democratic Republic of the

2. …Congo, ironically named the ‘capital of rape’ in the world,

This is not an exhaustive list, but examples of the type of language to edit.

• Language is too colloquyial, like a conversation or an informal write-up. These types of sentences are not appropriate in methods sections “Another issue discussed in multivariable logistic regression is the goodness-of-fit of the estimated models: To what extent do models significantly fit the data?”

• Similarly, “It is worthy mentioning that…”

Introduction

• The link between IPV and SDGs needs to be drawn out a little more before making this claim, “The high levels of IPV prevalence could be a strong impediment to achieving SDGs by 2030.”

• The rationale for why this study explores regional variation in DRC is not totally clear. Similarly, authors hypothesize that polygyny will be associated with IPV but also say this should be different across regions – why? Further, the rationale for a positive linkage between IPV and polygyny, which is mixed in other studies, is not well explained for their prediction in this cultural context.

Method

• The early information presented in the method actually belongs in the introduction on the cultural context for DRC, whereas factual info (e.g., literacy rates of each region sampled) should be in methods. This information still needs to be streamlined, such that only relevant parts that align with the argument presented in the introduction should go in.

• All major claims require robust citations. For example, this sentence is making a claim without citations, so the language needs to be tempered or substantiated with evidence: “However, a latent form of patriarchy still governs these provinces because men have more power than women; these latter have the role of procreation solely to ensure that the families have enough men and women.”

• The conceptualization of monogamy is not clear. Just because women denoted no other wives in the family, the unit was considered monogamous? This is not clear whether the expectation and general norm is for the husband to take more wives; please clarify this. It conflates monogamous marriage (by choice) with early marriage (whereby husbands are still able to/can choose to take other wives in the future).

• Information about other studies and any kins of literature review should not be in the method (e.g., about methodological choices in other studies, rationale for controls etc).

• Controls is not labeled right; do you mean covariates? Each exposure here should be a subsection of this section and explained properly.

• Analyses are missing many key details. What were the “spatial variations” used? This is confusing and unclear. Were all 26 districts analysed individually or at regional levels? Much more detail is required about analyses and how it was demarcated, how many N were in each region etc etc.

• It is not enough to report, “assumptions about logistic regression were carefully checked.” The assumptions are usually explained clearly in a paper.

Results

• Why are unadjusted odds ratios being presented rather than adjusted ratios?

Discussion

• What are the “community-specific effects of polygyny on IPV”? It is unclear what it meant by this and why community-level differences are important to look at. Similarly, variables that could vary by community were not analyzed in this paper, so it is unclear what the theory is that connects the choice of community-level unit to the research topic.

• Examples of interventions at regional levels, that are not being done, should also be in this section.

• This study used representative sample with survey data collected as part of a national study. However, it is not clear if the information of regional rates of IPV is new or not; presumably the survey authors have already published on IPV prevalence in these provinces? If so, is the novel contribution that this study is analyzing community/regional prevalence by marriage type? The fact of regional variance in itself is not a new finding. We would expect this at all regions in a subnational survey.

• Overall, these questions require much more clarification and the unique information contributed by this study to public health is needed.

---

## [Decision Letter · Decision Letter 1]

1 May 2024

PGPH-D-23-00057R1

Polygyny and intimate partner violence among married women: Sub-national estimates from a cross-sectional study in the Democratic Republic of the Congo

Dear Dr. Tsala Dimbuene,

Thank you for submitting your manuscript to PLOS Global Public Health. After careful consideration, we feel that it has merit but does not fully meet PLOS Global Public Health’s publication criteria as it currently stands. Therefore, we invite you to submit a revised version of the manuscript that addresses the points raised during the review process.

We look forward to receiving your revised manuscript.

Kind regards,

Ditte S Linde, Ph.D.

Academic Editor

Journal Requirements:

Additional Editor Comments (if provided):

Comments from review of manuscript PGPH-D-23-00057R1

• Line 26: Unless you state the year, use indicate, not indicated

• Line 27: Recently? State which regions have been having highest IPV and when SSA started recording the highest IPV (and reasons, if known)

• Lines 28-29 – why is knowing the association between IPV and polygyny important?

• Line 34 and throughout the paper, use of study a bit of a misnomer – that was analysis of DHS data

• Lines 35, 42, 43: background does not bring out the need for this analysis/association

• Lines 47-51, how does knowing the difference in just these 2 parameters contribute or add value?

• Lines 56-59: As presented, the only 2 policy actions would be targeting polygynous families with IPV interventions, or intervene to dismantle polygyny - the latter may not be an option; the former would be tricky and amount to labelling

• Lines 84-85 – a brief explanation of the absence of domestic violence would be helpful

• Line 89 – Peru province is in which country

• Line 91 – IPV prevalence is 30% in which country/region?

• Lines 99-101: while I agree with this, restricting to polygyny vs monogamy may not add as much value as assessing concentrations of IPV and explanatory factors that are amenable to interventions – see comments on lines 56-59

• Lines 130-131: Only parameter tested is polygyny - see earlier comment on opportunity for intervention in polygyny.

• Line 149: …men have more power? Which type? Physical? Economic?

• Line 150: This statement from the authors (not cited) is offensive: [women] have the role of procreation solely to ensure that the families have enough men and women

• Line 155 – be consistent in the use of polygyny vs. polygamy – the author seems to use the terms interchangeably in some places, including table 3

• Lines 158-159: add citation for these documents

• Line 171: citations are numbered – be consistent

• Line 173: what does ‘conceptions of sexuality’ mean?

• Line 178 – data source is 10 years old. Wasn’t there a more recent DHS?

• Line 198 – remove ‘the’ (the marriage)

• Lines 199, 209 – final sample is only women and eligibility includes men; questions on Table 1 imply only women were included in the analysis, so remove definition of eligibility for men

• Lines 228-239: Most of this should be in background, not methods

• Line 249 – final…..move to methods or results if they refer to this paper

• Lines 254-255 – for No – are all the 5 lumped together, or at least one, the way Yes is treated?

• Lines 290-291: Who made the decision to not seek guidance from ethics committee? The authors should have requested for a waiver from an in-country IRB on account of secondary data. By the way (to the editor), what's the journal's policy?

• Line 320: 43% of married women experience violence while prevalence of polygyny is less than half (19%) and now all have experience IPV – this implies other reasons/drivers are equally important and account for more than polygyny so should have strengthened this paper

• Line 338 – married polygynous or monogamous?

• Line 359 – spell out which co-variates were adjusted for?

• Line 369-370 – delete ‘married’ on both lines – redundant

• Line 394 – women’s and child’s – both either plural or singular

• Lines 394-395: How does knowing provincial variations add value

• Line 399: delete the sentence: Second, DRC is one of Central Africa’s countries. – does not add value

• Line 428: word missing between ‘representative’ and ‘to’

• Lines 429-230: This sentence is not clear

• Lines 432-433: what conflates monogamy with early marriage? This means there should have been adjustment for age - it's not clear what was included in the adjustment presented in Table 4

• Line 433-435: Second…… it's not clear what is being communicated here

• Line 435-436 – Third…… was this in the analysis and results? I can't find reference to husband’s characteristics

• Lines 437-438 and 428-439 - combine the 2 sentences in each

• Lines 449-452: it's not clear how knowing that IPV and polygyny can inform policy? What interventions are you having in mind? See earlier comment on Lines 56-59

Reviewers' comments:

Reviewer's Responses to Questions

**Comments to the Author**

1. If the authors have adequately addressed your comments raised in a previous round of review and you feel that this manuscript is now acceptable for publication, you may indicate that here to bypass the “Comments to the Author” section, enter your conflict of interest statement in the “Confidential to Editor” section, and submit your "Accept" recommendation.

Reviewer #2: All comments have been addressed

2. Does this manuscript meet PLOS Global Public Health’s publication criteria? Is the manuscript technically sound, and do the data support the conclusions? The manuscript must describe methodologically and ethically rigorous research with conclusions that are appropriately drawn based on the data presented.

Reviewer #2: Partly

3. Has the statistical analysis been performed appropriately and rigorously?

Reviewer #2: Yes

4. Have the authors made all data underlying the findings in their manuscript fully available (please refer to the Data Availability Statement at the start of the manuscript PDF file)?

Reviewer #2: Yes

5. Is the manuscript presented in an intelligible fashion and written in standard English?

Reviewer #2: Yes

6. Review Comments to the Author

Reviewer #2: Comments are attached

7. PLOS authors have the option to publish the peer review history of their article (what does this mean?). If published, this will include your full peer review and any attached files.

**Do you want your identity to be public for this peer review?** For information about this choice, including consent withdrawal, please see our Privacy Policy.

Reviewer #2: No

---

## [Decision Letter · Decision Letter 2]

17 Sep 2024

PGPH-D-23-00057R2

Polygyny and intimate partner violence among married women: Sub-national estimates from a cross-sectional study in the Democratic Republic of the Congo

Dear Dr. Tsala Dimbuene,

Thank you for submitting your manuscript to PLOS Global Public Health. After careful consideration, we feel that it has merit but does not fully meet PLOS Global Public Health’s publication criteria as it currently stands. Therefore, we invite you to submit a revised version of the manuscript that addresses the points raised during the review process.

The manuscript has been evaluated by four reviewers, and their comments are available below, and in the attached files.

Although the reviewers agree that the topic is important, they each have a list of minor concerns that need to be addressed in order to improve the detail and clarity of your paper.

In addition, I would like to request more detail regarding the analysis reported in table 3. Earlier in the manuscript (in the methods section), you mention that "Statistical significance was tested at national and sub-national levels using Pearson’s chi-square tests", but there is no description of the tests performed in the results section. I assume that the p values reported in table 3 are from chi-square analyses, but I cannot be sure. Also, I can see how the association between province and each of the four dependent variables can be assessed using chi-square (i.e., the bolded p values at the bottom of the table), but it is not clear how the p values for each province were derived. For example, you report a non-significant association between marital status and physical violence in Kinshasa: Is this a test comparing the score of 13.8 and the score of 25.7? What test was used? Or was this a test using the underlying data rather than the means? Also, I believe that the numbers reported in the table are percentages, but this information is not stated. In short, please provide a lot more information about how the data were analysed.

Could you please revise the manuscript to carefully address the concerns raised?

A rebuttal letter that responds to each point raised by the editor and reviewer(s). You should upload this letter as a separate file labeled 'Response to Reviewers'.

We look forward to receiving your revised manuscript.

Kind regards,

Steve Zimmerman, PhD

PLOS Staff Editor

Additional Editor Comments (if provided):

Reviewers' comments:

Reviewer's Responses to Questions

**Comments to the Author**

1. If the authors have adequately addressed your comments raised in a previous round of review and you feel that this manuscript is now acceptable for publication, you may indicate that here to bypass the “Comments to the Author” section, enter your conflict of interest statement in the “Confidential to Editor” section, and submit your "Accept" recommendation.

Reviewer #2: (No Response)

Reviewer #3: All comments have been addressed

Reviewer #4: (No Response)

Reviewer #5: (No Response)

2. Does this manuscript meet PLOS Global Public Health’s publication criteria? Is the manuscript technically sound, and do the data support the conclusions? The manuscript must describe methodologically and ethically rigorous research with conclusions that are appropriately drawn based on the data presented.

Reviewer #2: Partly

Reviewer #3: Yes

Reviewer #4: (No Response)

Reviewer #5: Yes

3. Has the statistical analysis been performed appropriately and rigorously?

Reviewer #2: I don't know

Reviewer #3: Yes

Reviewer #4: (No Response)

Reviewer #5: Yes

4. Have the authors made all data underlying the findings in their manuscript fully available (please refer to the Data Availability Statement at the start of the manuscript PDF file)?

Reviewer #2: Yes

Reviewer #3: Yes

Reviewer #4: (No Response)

Reviewer #5: Yes

5. Is the manuscript presented in an intelligible fashion and written in standard English?

Reviewer #2: Yes

Reviewer #3: Yes

Reviewer #4: (No Response)

Reviewer #5: Yes

6. Review Comments to the Author

Reviewer #2: I am unable to respond to availability of data (no 4 above) so response is place holder. Maybe 'don't know' option should be included for reviewers. Comments are as below:

Re-review of PGPH-D-23-00057R2 (Polygyny and intimate partner violence among married women: Sub-national estimates from a cross-sectional study in the Democratic Republic of the Congo)

Grateful to the authors for addressing most of my previous concerns raised in the first review except in one or 2 places (see comments). I have very few, very minor, comments.

Abstract:

Line 45: ‘components’ – consider replacing with ‘manifestations’ or some other term

Introduction:

Lines 93-94: The difference between 36% and 30% is small so phrases like ‘as high as’ or the ‘high levels of IPV’ not supported by provided data. Perhaps tone down the wording. Also term ‘drastic’ on line 95 – too strong

Line 104 – ‘to’ polygyny

Line 106: ‘To provide with insights on IPV’ not clear what is meant – maybe remove ‘with’?

107-108 – cite the framework immediately after, separate from its description

117 – not necessarily ‘many’; more than one wife is enough to qualify as polygynous

Methods:

150-151: Except for the province of Kinshasa, the Capital City of the country, the DRC is

subdivided into matrilineal societies (South Bandundu and Kongo central)….sentence sounds incomplete.

160-176 – more like background material – same comment made at first review

Line 180 – please provide citation for the DHS. Same comment made at initial review

Line 206 – ‘component’ – comment as previous

222-225: reads like background info

Results:

Line 321 – respectively does not apply

Reviewer #3: This manuscript reported Polygyny and IPV among married women in the Democratic Republic of the Congo. This study brings some essential facts, however, there are some points I would like the authors to address to further strengthen the manuscript.

Abstract

1. The authors are suggested to kindly maintain one form of tense in the background.

2. Can the authors clarify, what they want to mean by differentials in the conclusion?

3. Is there any policy introduced by the government to lessen IPV among women in DRC? If yes, then the authors are requested to mention those.

Introduction

4. Line 77, when it has already been stated in SDG, there is no need to put for instance.

5. Lines 78-80, the authors are requested to put intext citations, since, its stating that previous studies have concluded.

6. Line 86, can the authors specific what they want to intend to the readers by small scale societies?

7. Line 92, authors can do away writing, “for example”, since you are stating a fact, there is no need of writing this as an example.

8. Line 98, The regions referred here, is it the study region?

9. The overall introduction provides some missing links to the readers. For e.g. The 3rd and 4th paragraph of page 5, do not show any interconnection. I rather suggest the authors, to kindly arrange the paragraphs of the introduction in a more story telling manner, which will help the readers to understand better.

10. Authors are requested to kindly provide with intext citations in Line 118-120.

11. Line 124, do the authors mean to say, that the study is based on data collected from 16 countries, to show inequality among IPV within the democratic republic of Congo? Please clarify.

12. I think, it will be beneficial for the readers if the authors clearly mention the research/study gap of this study?

Data

13. I think the authors can think about putting study setting in the data section, as this paragraph does not imply to data and data related information of the study. The study setting is explaining IPV in different regions of DRC.

14. Line 232 and 233 are inferring to same meaning. Can the authors kindly write about the level of education in one sentence to give a complete meaning?

15. Line 240, Can the authors clarify what they intend to signify by stating that it has been published elsewhere? They are requested to give a complete information.

16. Can the author mention the value of VIF, while stating no multicollinearity existed? Also, for goodness of fit of the models, can the authors mention the value of R2?

17. Can the authors provide a table showing the prevalence of all independent and control variables of the study population? For e.g. Prevalence rates of poorest, poorer, richer, and richest among the study population?

Discussion

18. I found the discussion of the study as a reiteration of same facts. I would rather suggest, the authors if they incorporate the associations of other variables such as level of education, poverty with IPV along with polygyny, the study will give a better read.

Reviewer #4: (No Response)

Reviewer #5: The manuscript titled "Polygyny and Intimate Partner Violence Among Married Women: Sub-National Estimates from a Cross-Sectional Study in the Democratic Republic of the Congo" explores the relationship between polygyny and intimate partner violence (IPV) among married women. The study utilizes data from the Demographic and Health Survey (DHS) 2013-2014. The study aims to provide a detailed understanding of how polygyny correlates with various forms of IPV—physical, emotional, and sexual—across different provinces in the DRC. The manuscript claims that women in polygynous unions are at a higher risk of experiencing IPV compared to those in monogamous marriages. Additionally, it highlights significant spatial variations in the prevalence of both polygyny and IPV, suggesting the need for region-specific interventions.

Strengths

Relevance and Contribution: The manuscript addresses a pressing public health and social issue in the DRC, contributing valuable insights into the complex dynamics between marital structure and IPV. The focus on sub-national estimates is particularly commendable, as it uncovers important regional disparities that can inform targeted policy responses.

Methodological Approach: The use of DHS data and multivariate logistic regression to analyze the relationship between polygyny and IPV provides a robust and credible foundation for the study. The data analysis is well-executed, with the clear and coherent presentation of results, supported by appropriately detailed tables and figures.

Clarity and Organization: The manuscript is generally well-organized, with a logical flow that facilitates understanding. The results are presented clearly, making it easy to follow the narrative from the introduction through to the discussion.

Weaknesses

Contextual Gaps: The introduction would benefit from a more comprehensive contextualization of the socio-cultural factors that contribute to the prevalence of polygyny in the DRC.

Insufficient Exploration of Regional Variations: While the study provides important data on regional variations, the discussion section could be strengthened by delving deeper into the potential causes of these differences. Providing a more nuanced analysis of why certain regions exhibit higher rates of polygyny and IPV would enhance the manuscript's overall impact.

Policy Implications: The discussion of policy implications is somewhat underdeveloped. Given the strong association between polygyny and IPV, the manuscript could benefit from more concrete recommendations for interventions or policies aimed at reducing IPV in regions with high polygyny rates.

Limitations and Specificity: The limitations section could be expanded to include a more detailed discussion of the challenges associated with using cross-sectional data, particularly regarding the potential for underreporting of IPV. Moreover, the manuscript would benefit from a clearer articulation of the specific gaps in the literature that this study aims to fill.

Recommended Course of Action

The manuscript is a valuable contribution to the literature on IPV and polygyny, particularly within the context of the DRC. However, I recommend revisions to address the contextual and theoretical gaps, explore regional variations more thoroughly, and strengthen the discussion of policy implications. Expanding the limitations section and clarifying the study's specific contributions to the existing literature would also be beneficial.

Abstract

• Lines 39: Specify the putative factors adjusted for in the analysis to provide clarity.

• Lines 49-50: Simplify the language by specifying the nature of the variations, e.g., "Kasai Occidental had higher rates of IPV."

• Lines 56-57, 442-443: Summarize the key findings more explicitly in the conclusion, mentioning which provinces had the highest rates of IPV and the overall impact of polygyny.

• Lines 58-59: Mention specific interventions or policy changes that could strengthen the impact of the conclusion.

Introduction

• Line 73: Use "predominantly directed at" instead of "mostly against."

• Line 74: Replace "attention in" with "attention over."

• Lines 75-78: Merge these lines into one sentence for clarity, transitioning from general to specific (SDG to SDG 5.2).

• Line 90: Use "Data suggests" instead of "available data suggests."

• Line 93: Replace "overall and worldwide" with more comprehensive terms like "the global prevalence of IPV."

• Lines 89-94: Reorganize for a more logical flow—start with the global context, move to regional specifics, then discuss the DRC.

• Line 94: Use "these high levels" instead of "the high levels."

• Line 95: Replace "unless drastic measures" with a more prescriptive phrase like "necessitating urgent action."

• Line 106: Instead of "the researcher in the context," use "the research context," focusing on how research has explored various drivers of the issue.

• Line 118: Include specific findings from key studies, such as how much more prevalent IPV is in polygynous unions compared to monogamous ones.

• Lines 126-129: Explicitly state the gaps in the literature that this study aims to fill.

• Line 129: Consider stating "To investigate the homogeneity or heterogeneity..."

• Line 132: Use "expected to provide" instead of "provide."

Methods

• Lines 145-147: To strengthen the theoretical framework, include information on how patriarchal norms differ in matrilineal vs. patrilineal societies and how these structures influence polygyny and IPV prevalence.

• Line 157: Use "into the mapping" instead of "on the mapping."

• Line 265: Include details on how missing data was handled, particularly regarding IPV reporting, which often suffers from underreporting.

Results

• Line 301: Given the focus on polygyny and IPV, including the prevalence of polygyny and the sample size of provinces in Table 2 would be more comprehensive and supplement Figure 1.

• Consider separating the description of prevalence rates and spatial variations of IPV into distinct subsections for clarity.

• Lines 339-346: Explore the reasons behind these variations to add depth to the analysis.

• Lines 379-384: Provide a more detailed explanation of the implications of the AORs presented in Table 4, particularly for provinces like Kongo Central, where the AOR was marginally significant. Clarify whether these findings suggest a trend or require cautious interpretation due to wide confidence intervals.

Discussion

• Lines 418-419: Expand the discussion on the policy implications of the findings, particularly how understanding sub-national variations can inform targeted interventions. Discuss specific strategies that could be implemented at the provincial level to address IPV in regions with high polygyny rates.

Conclusion

• Line 442: Use "show" or "reveal" instead of "showed."

• Lines 448-452: Strengthen the conclusion by incorporating specific recommendations for actionable steps, acknowledging potential limitations, and emphasizing the need for localized interventions. Suggest that future research explore additional factors, such as male partner characteristics and community-level influences, to deepen our understanding of IPV drivers in different contexts.

Other Suggestions

• Improve the clarity and comprehensiveness of table and figure labels.

• Provide a clear and operational definition of IPV specific to this study's universe.

• Language editing is required.

7. PLOS authors have the option to publish the peer review history of their article (what does this mean?). If published, this will include your full peer review and any attached files.

**Do you want your identity to be public for this peer review?** For information about this choice, including consent withdrawal, please see our Privacy Policy.

Reviewer #2: No

Reviewer #3: No

Reviewer #4: **Yes: **Anshika Singh

Reviewer #5: **Yes: **Shubhi Yadav

---

## [Decision Letter · Decision Letter 3]

24 Oct 2024

PGPH-D-23-00057R3

Polygyny and intimate partner violence among married women: Sub-national estimates from a cross-sectional study in the Democratic Republic of the Congo

Dear Dr. Tsala Dimbuene,

Thank you for submitting your manuscript to PLOS Global Public Health. After careful consideration, we feel that it has merit but does not fully meet PLOS Global Public Health’s publication criteria as it currently stands. Therefore, we invite you to submit a revised version of the manuscript that addresses the points raised during the review process.

Whilst the reviewer comments have been accessed adequately, we note that the previous editorial comments were not addressed. These comments, along with some additional minor requests, are copied below. Please ensure you provide your responses as part of a revision.

We look forward to receiving your revised manuscript.

Kind regards,

Marianne Clemence

Staff Editor

Additional Editor Comments (if provided):

1. I would like to request more detail regarding the analysis reported in table 3. Earlier in the manuscript (in the methods section), you mention that "Statistical significance was tested at national and sub-national levels using Pearson’s chi-square tests", but there is no description of the tests performed in the results section. I assume that the p values reported in table 3 are from chi-square analyses, but I cannot be sure. Also, I can see how the association between province and each of the four dependent variables can be assessed using chi-square (i.e., the bolded p values at the bottom of the table), but it is not clear how the p values for each province were derived. For example, you report a non-significant association between marital status and physical violence in Kinshasa: Is this a test comparing the score of 13.8 and the score of 25.7? What test was used? Or was this a test using the underlying data rather than the means? Also, I believe that the numbers reported in the table are percentages, but this information is not stated. In short, please provide a lot more information about how the data were analysed.

Please review our guidelines for statistical reporting and ensure your manuscript complies with these (https://journals.plos.org/globalpublichealth/s/submission-guidelines#loc-statistical-reporting)

2. The data used to inform this study are over 10 years old, and may not be relevant to a contemporary context. Please be sure to discuss this as a limitation.

3. Please remove the final sentence of your conclusion, which is not directly relevant to the results presented here.

Reviewers' comments:

Reviewer's Responses to Questions

**Comments to the Author**

1. If the authors have adequately addressed your comments raised in a previous round of review and you feel that this manuscript is now acceptable for publication, you may indicate that here to bypass the “Comments to the Author” section, enter your conflict of interest statement in the “Confidential to Editor” section, and submit your "Accept" recommendation.

Reviewer #4: All comments have been addressed

Reviewer #5: All comments have been addressed

2. Does this manuscript meet PLOS Global Public Health’s publication criteria? Is the manuscript technically sound, and do the data support the conclusions? The manuscript must describe methodologically and ethically rigorous research with conclusions that are appropriately drawn based on the data presented.

Reviewer #4: Partly

Reviewer #5: Yes

3. Has the statistical analysis been performed appropriately and rigorously?

Reviewer #4: Yes

Reviewer #5: Yes

4. Have the authors made all data underlying the findings in their manuscript fully available (please refer to the Data Availability Statement at the start of the manuscript PDF file)?

Reviewer #4: Yes

Reviewer #5: Yes

5. Is the manuscript presented in an intelligible fashion and written in standard English?

Reviewer #4: Yes

Reviewer #5: Yes

6. Review Comments to the Author

Reviewer #4: (No Response)

Reviewer #5: Thank you for submitting the revised version of the manuscript. I appreciate the thoughtful and comprehensive revisions you have made in response to the initial review. The revisions have substantially improved the manuscript, and it is now much clearer, more comprehensive, and actionable. You have effectively addressed the key concerns, and I believe this study will make a valuable contribution to the literature on polygyny and IPV in the DRC.

I look forward to seeing the final version published.

Thank you for your diligent work on this important issue.

7. PLOS authors have the option to publish the peer review history of their article (what does this mean?). If published, this will include your full peer review and any attached files.

**Do you want your identity to be public for this peer review?** For information about this choice, including consent withdrawal, please see our Privacy Policy.

Reviewer #4: No

Reviewer #5: **Yes: **Shubhi Yadav

---

## [Decision Letter · Decision Letter 4]

13 Dec 2024

Polygyny and intimate partner violence among married women: Sub-national estimates from a cross-sectional study in the Democratic Republic of the Congo

PGPH-D-23-00057R4

Dear Prof. Tsala Dimbuene,

We are pleased to inform you that your manuscript 'Polygyny and intimate partner violence among married women: Sub-national estimates from a cross-sectional study in the Democratic Republic of the Congo' has been provisionally accepted for publication in PLOS Global Public Health.

Best regards,

Zulma Vanessa Rueda, M.D. Ph.D.

Academic Editor

The authors have addressed all reviewers comments. The topic is very relevant, and I think during the formatting of the accepted paper, the authors can include the limitation that the reviewer suggested.

Thanks for considering the journal for your manuscript.

Reviewer Comments (if any, and for reference):

Reviewer's Responses to Questions

**Comments to the Author**

1. If the authors have adequately addressed your comments raised in a previous round of review and you feel that this manuscript is now acceptable for publication, you may indicate that here to bypass the “Comments to the Author” section, enter your conflict of interest statement in the “Confidential to Editor” section, and submit your "Accept" recommendation.

Reviewer #5: All comments have been addressed

Reviewer #6: (No Response)

2. Does this manuscript meet PLOS Global Public Health’s publication criteria? Is the manuscript technically sound, and do the data support the conclusions? The manuscript must describe methodologically and ethically rigorous research with conclusions that are appropriately drawn based on the data presented.

Reviewer #5: Yes

Reviewer #6: Yes

3. Has the statistical analysis been performed appropriately and rigorously?

Reviewer #5: Yes

Reviewer #6: I don't know

4. Have the authors made all data underlying the findings in their manuscript fully available (please refer to the Data Availability Statement at the start of the manuscript PDF file)?

Reviewer #5: Yes

Reviewer #6: Yes

5. Is the manuscript presented in an intelligible fashion and written in standard English?

Reviewer #5: Yes

Reviewer #6: Yes

6. Review Comments to the Author

Reviewer #5: (No Response)

Reviewer #6: I think the authors have satisfactorily addressed the previous reviewer comments. Because the data are 10 or more years old, the authors could state in the methods that 2013-2014 data are the most current DHS data available, but also mention this as a limitation (as suggested by a past reviewer).

7. PLOS authors have the option to publish the peer review history of their article (what does this mean?). If published, this will include your full peer review and any attached files.

**Do you want your identity to be public for this peer review?** For information about this choice, including consent withdrawal, please see our Privacy Policy.

Reviewer #5: **Yes: **Shubhi Yadav

Reviewer #6: **Yes: **M J Haworth-Brockman
